# Trends in healthcare use in children aged less than 15 years: a population-based cohort study in England from 2007 to 2017

Judith Ruzangi ![ORCID], Mitch Blair, Elizabeth Cecil, Geva Greenfield, Alex Bottle, Dougal S Hargreaves, Sonia Saxena ![ORCID]

Department of Primary Care & Public Health, Imperial College London, London, UK

**Correspondence to**
Judith Ruzangi;
j.ruzangi@imperial.ac.uk

## ABSTRACT

**Objective** To describe changing use of primary care in relation to use of urgent care and planned hospital services by children aged less than 15 years in England in the decade following major primary care reforms from 2007 to 2017

**Design** Population-based retrospective cohort study.

**Methods** We used linked data from the Clinical Practice Research Datalink to study children's primary care consultations and use of hospital care including emergency department (ED) visits, emergency and elective admissions to hospital and outpatient visits to specialists.

**Results** Between 1 April 2007 and 31 March 2017, there were 7 604 024 general practitioner (GP) consultations, 981 684 ED visits, 287 719 emergency hospital admissions, 2 253 533 outpatient visits and 194 034 elective admissions among 1 484 455 children aged less than 15 years. Age-standardised GP consultation rates fell (−1.0%/year) to 1864 per 1000 child-years in 2017 in all age bands except infants rising by 1%/year to 6722 per 1000/child-years in 2017. ED visit rates increased by 1.6%/year to 369 per 1000 child-years in 2017, with steeper rises of 3.9%/year in infants (780 per 1000 child-years in 2017). Emergency hospital admission rates rose steadily by 3%/year to 86 per 1000 child-years and outpatient visit rates rose to 724 per 1000 child-years in 2017.

**Conclusions** Over the past decade since National Health Service primary care reforms, GP consultation rates have fallen for all children, except for infants. Children's use of hospital urgent and outpatient care has risen in all ages, especially infants. These changes signify the need for better access and provision of specialist and community-based support for families with young children.

## BACKGROUND

Primary care is at the heart of effective health systems but globally both primary care and hospital services are increasingly under pressure.[1] The strengths of primary care include comprehensive, coordinated, preventive and planned care that has been shown to improve population health and reduce avoidable health system waste.[2] In the UK National

### Strengths and limitations of this study

► This is the largest nationally representative population-based study to date examining trends in children's healthcare use in the National Health Service in England.
► The cohort design using individual-level linked data from primary care and hospitals enabled a whole systems analysis of children's healthcare use.
► Previous studies focused on children's primary care use have not previously been reported.
► Improvements in data quality have enabled analysis of children's emergency department visit and outpatient visits.
► The study population was children registered with a general practice, which underrepresents those from poorer postcodes, mobile or homeless populations.

Health Service (NHS), over 98% of children are registered from birth with a family physician (known as a general practitioner (GP)) and receive preventive care including vaccinations and development checks as well as first contact care for acute problems.[3] Infants and preschool children have among the highest visit rates of any age group to primary care.[1] GPs also diagnose and manage chronic illnesses that emerge across the life course and play a crucial role in prioritising onwards referral for specialist care in hospital and outpatient settings. This cradle to grave model has been in place since the creation of the NHS over 70 years ago. However, major reforms in NHS primary care over the past decade are thought to have pushed clinical and administrative GP workloads to saturation point and reduced children's access to GPs in and out of hours.[4–6] From 2004, most GPs opted out of providing out of hour's services and took on a greater role in active monitoring and management of a rising population burden of chronic conditions in

**Box 1 Summary of major UK National Health Service reforms 2004 and key definitions**

**Primary care reforms**[4]
Allowed primary care physicians (general practitioners) greater flexibility in the services they provided, for example, opting out of responsibility for out-of-hours care.
Financial incentive scheme for primary care physicians to deliver clinical and organisational care, assessed through performance target achievement (the Quality and Outcomes Framework).
**Emergency care reforms**
Tightening of targets so that 98% of patients wait no more than 4 hours in an emergency department (ED) from arrival to hospital admission, transfer or discharge.[35]
**Hospital reforms**
Introduction of payment-by-results schemes changed the way a hospital was paid, from a block contract payment system for service provision to one remunerating activity, such as episodes of care.[36]
**General practitioner consultation**—a consultation with a local family physician who delivers preventive care including vaccinations and development checks as well as first contact care for acute problems.
**Outpatient visit**—a planned appointment with a hospital specialist (eg, a general paediatrician or a subspecialist such as a paediatric neurologist or paediatric gastroenterologist).
**Urgent care**—unplanned immediate care received in EDs including urgent care centres that are commonly colocated within English EDs.

adults, driven by financial incentive schemes (box 1).[7] Face to face GP consulting rates increased by 10% and telephone consulting rates doubled in the decade to 2013/2014.[1]

There is some evidence that children's use of hospital emergency services is sensitive to access and availability of primary care. In the period following primary care reforms of 2004, there were large increases in children's visits to emergency departments (EDs) and rises in short-stay hospital admissions for primary care sensitive conditions.[7] A third of all British children aged less than 15 years currently visit ED each year.[8] This figure rises to 40% in practices where access is poor.[3] Two-thirds of children visit outside of normal GP consulting hours although the peak time for ED visits is between 16:00 and 18:00.[3] A relatively high proportion of children visiting ED (15%–80%) is discharged with no treatment or present with conditions that could potentially have been treated in primary care.[9 10] Previous ecological studies have been unable to provide population-based estimates of children's health service using individual-level data across the primary and secondary care interface. The aim of this study was to describe trends in use of primary care in relation to use of urgent care and planned NHS hospital services by children aged less than 15 years in England in the decade following major primary care reforms.

## METHODS
### Study design and data sources
We carried out a cohort study using prospectively collected data from the Clinical Practice Research Datalink (CPRD),

which is the largest validated primary care research database within the UK.[11] It contains longitudinal, patient-level, anonymised computerised health records from more than 600 participating UK general practices for 22 million patients.[11] While only representing 8% of the UK population, age, sex and ethnicity reflect demographics of the UK.[11 12] We used CPRD linked to Hospital Episodes Statistics (HES) to obtain information on ED visits, emergency hospital admissions, elective admissions and outpatient attendance. HES contains information on NHS hospital activity in England.

### Population
We constructed an open cohort including all children aged less than 15 years in HES-linked CPRD registered 'up to standard' CPRD-participating general practices during the study period from 1 April 2007 to 31 July 2017. Each child contributed to the time of observation from birth or first registration date. Follow-up for each year was continuous for each child until either the date they transferred out of practice, reached the age of 15 years, died or reached the end of the study period (31 July 2017), whichever came first.

Children were assigned to one of four developmental age groups; infants aged <1 year, preschool children aged 1–4 years, school children aged 5–9 years and teenagers aged 10–14 years. Since CPRD data do not contain birthdates, we ascribed a birthdate of the first of the recorded month and year of birth to each child. For children whose record had missing data for the month of their birth, we assumed a birthdate as 1 January on their year of birth. Children contributed data on health service use at different ages across each year of the study period. For each financial year, the age for each child was defined as the number of years since the birthdate of that child to the end of that financial year (31 March). Hence, this was an open cohort with children entering and leaving the sample as well as changes in age across the decade.

### Outcomes
Our main outcomes were age-standardised rates for GP consultation, ED visits, emergency and elective hospital admissions and outpatient visits. We defined a GP consultation as any face to face consultation for illness that took place on practice premises, excluding consultations for routine preventive care (immunisation and development checks). An ED visit was defined as an ED attendance at a consultant-led ED with 24 hours service, full resuscitation facilities and designated accommodation for the reception of accident & emergency (A&E) patients and into other types of A&E/minor injury department. We excluded visits to consultant-led mono speciality A&E service such as specialist emergency eye units and NHS walk-in centres. Emergency admissions were admissions that were 'unpredictable and occur at short notice because of clinical need', whereas elective admissions were defined as occurring when the 'decision to admit can be separated in time from the actual admission'. We defined

an outpatient attendance where a child was recorded as having been seen by the intended care professional on the date of appointment on the HES outpatient appointment data set.[13] These outcomes are described in online supplementary file 1.

## Analysis

We calculated annual rates per 1000 child-years for each outcome for each financial year by summing the total number of events divided by the total child-years of observation. To enable comparison across years, rates were age standardised to the 2016 mid-year English population aged <15 years.[14] Similarly, annual rates were calculated for each developmental age group from 2007 to 2017. However, due to poor HES A&E data quality before 2011, we only report ED visit rates from 2011/2012.[15] We calculated the change from baseline rate in 2007/2008 to 2016/2017, except for ED visits where we used a baseline reference year of 2011/2012. We calculated the conversion proportion for emergency admission from 2011/2012 as the proportion of ED visits that led to emergency admissions.

## Patient and public involvement

Patients and the public were not directly involved in writing this paper.

## RESULTS

Overall, 1 484 455 children from 408 practices contributed to this 10-year cohort study.

From 1 April 2007 to 31 March 2017, there were 7 604 024 GP consultations, 981 684 ED visits, 287 719 emergency admissions, 194 034 elective admissions, 2 253 533 outpatient visits among children aged less than 15 years in our population.

## GP consultations

The age-standardised GP consultation rate dropped from 2005 to 1864 per 1000 child-years between 2007/2008 and 2016/2017 (figure 1). In 2016/2017, the telephone contact rates were 0.1 per child and home visits 1 per 1000 children. The GP consultation rate in infants increased by 1%/year from 6084 per 1000 child-years in 2007/2008. The GP consultation rate in preschool children and those aged 5–9 years fell from 2707 to 2368 per 1000 child-years and 1416 to 1217 per 1000 child-years in the same study period, respectively. For children aged above 10 years, the rate was stable throughout the decade (1100 per 1000 child-years) (table 1).

## Urgent and emergency care; ED visits and emergency hospital admissions

From 2011 to 2017, the age-standardised ED visit rate increased by 1.6%/year. There was a much larger increase in 3.9%/year in annual ED visit rate for infants compared with older children. Since 2011, every year, 22% of ED visits led to an emergency admission. From 2007/2008 to 2016/2017, the age-standardised emergency admission

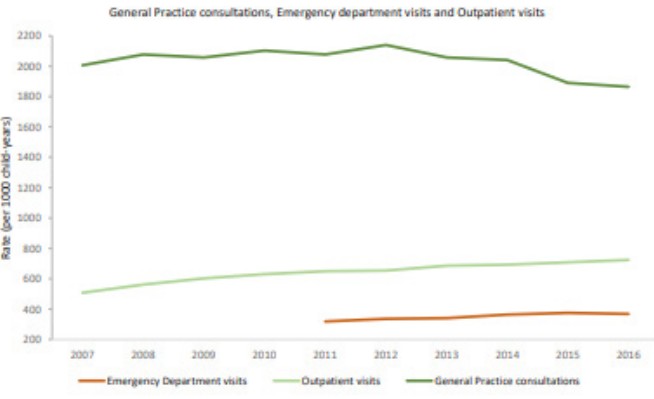

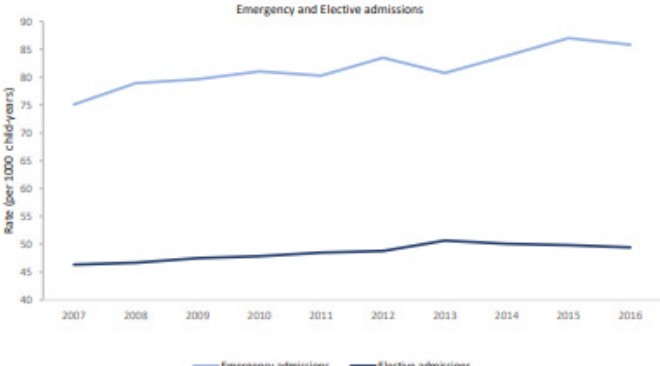

Rates were directly age-standardised to the 2016 mid-year English population

**Figure 1** Healthcare use in children aged less than 15 years.

rate increased by 1.5%/year from 75 per 1000 child-years and for infants by 3%/year from 312 per 1000 child-years. However, the emergency admission rate in preschool children and those aged above 5 years was stable throughout the decade (100–130 and <50 per 1000 child-years, respectively).

## Planned care; outpatient visits and elective admissions

From 2007/2008 to 2016/2017, the annual age-standardised outpatient visit rate increased from 508 to 724 per 1000 child-years (figure 1). The outpatient visit rate in infants increased by 7.3%/year over the study period. Rates in preschool children and those aged above 5 years also increased steadily throughout the same period from 500 to 700 per 1000 child-years (figure 2). Age-standardised elective admission rates over the same study period were stable from 46 to 49 per 1000 child-years but increased by 4.6%/year for infants and by 2%/year for teenagers. Rates were stable for preschoolers and school children (~50 per 1000 child-years).

## DISCUSSION
### Key findings

Infant contact rates with GPs have increased year on year over the past decade following major reforms in UK primary care but older children are using primary care less. GP consultations are the major point of contact with healthcare services across all age groups. Children's use of

**Table 1** Annual rates of healthcare use per 1000 child-years and relative change from baseline 2007 to 2017*,†

| | Infants <1 year | | | Preschool children 1–4 years old | | | School children 5–9 years old | | | Teenagers 10–14 years old | | |
|---|---|---|---|---|---|---|---|---|---|---|---|---|
| | Baseline | 2017 | % change* (95% CI) | Baseline | 2017 | % change* (95% CI) | Baseline | 2017 | % change* (95% CI) | Baseline | 2017 | % change* (95% CI) |
| General practice consultations | 6084 | 6722 | 10.5 (8.9 to 12.0) | 2707 | 2368 | −12.5 (−13.2 to 11.8) | 1416 | 1218 | −14.0 (−14.8, to 13.2) | 1143 | 1074 | −6.0 (−7.0 to 5.0) |
| Outpatient visits | 883 | 1529 | 73.1 (66.0 to 80.2) | 518 | 706 | 36.3 (33.7 to 38.9) | 477 | 649 | 36.2 (33.8 to 38.6) | 451 | 648 | 43.9 (41.1 to 46.7) |
| Emergency department visits† | 562 | 780 | 38.6 (32.8 to 44.5) | 419 | 497 | 18.8 (17.0 to 20.5) | 226 | 249 | 10.3 (8.2, 12.3) | 280 | 297 | 6.2 (4.2 to 8.1) |
| Emergency admissions | 312 | 410 | 31.6 (24.2 to 39.1) | 109 | 123 | 12.8 (8.8 to 16.8) | 38 | 39 | 2.8 (−2.9 to 8.5) | 34 | 34 | −1.4 (−8.1 to 5.2) |
| Elective admissions | 47 | 68 | 46.4 (14.1 to 78.6) | 53 | 52 | −3.0 (−13.7 to 7.6) | 48 | 47 | −2.4 (−10.3 to 5.5) | 38 | 46 | 20.7 (−1.0 to 42.3) |

*% change calculated as (rate in 2017 − rate in baseline)/rate in baseline; 95% CI.
†Baseline year used was 2007/2008 for all outcomes except emergency department visits, which was 2011/2012 due to poor data quality prior to this time.

urgent care in hospitals, including visits to EDs and emergency admission rates, has increased particularly among infants. There have been sharp increases in planned care including outpatient activity and admissions for elective care in infants and teenage children.

## Comparison with previous studies

Health system failures to provide responsive care in the community to meet the needs of the children and families might well explain the rise in urgent care use. Parents prefer using their regular general GP when their child is unwell, but they choose to visit the ED if they perceive their child's condition as serious or cannot access primary care.[16–18] It is estimated that 10% of infants attending the ED have no underlying medical problem,[19] yet this does not mean that 90% need to seek emergency care if they require support to care for their child. The 2004 primary care healthcare reforms focused heavily on improving chronic disease management in adults with long-term conditions and allowed primary care physicians to opt out of providing acute primary care services during evenings and weekends. This reduced access that may explain rising ED visits.[3] The expansion in the use of urgent care is consistent with trends in the UK and other high-income country settings where emergency admission rates in both children and adults have been rising for several years.[7 20]

We have previously reported GP consultation rates for illness in children of a similar magnitude for a CPRD cohort born in 2000 and another study of healthcare use in primary care study that is remarkably consistent with our findings here for children.[1 21] Infants, however, seem to be seeing their GP more frequently in 2017 compared with a decade ago. There is little research on children's outpatient service use and elective admissions but is consistent with overall rises in the total number of elective admissions, referrals to outpatient services and total outpatient attendances in English hospitals, which has increased markedly since 2007.[22–24]

Rising needs in the child population and demand from families are most likely contributors to the patterns of overall increasing healthcare use by children seen here.[8] Preterm birth is increasing and is the single biggest cause of neonatal morbidity and mortality in the UK and other countries where survival rates are increasing.[25] Thus, rising emergency admissions in infants could be explained by a growing morbidity burden in infants compared with other children. Recurrent admissions in children and young people with chronic conditions contribute substantially to total emergency admissions.[26] A report projecting the State of Child Health in 2030 concluded that Britain's children who make up nearly 20% of the population have a rising health burden and poorer health compared with other wealthy countries. Even more concerning is a widening in the gap in health between wealthy and poor children.[27] Hence growing morbidity burden particularly in deprived groups may be contributing to the rise in healthcare use.

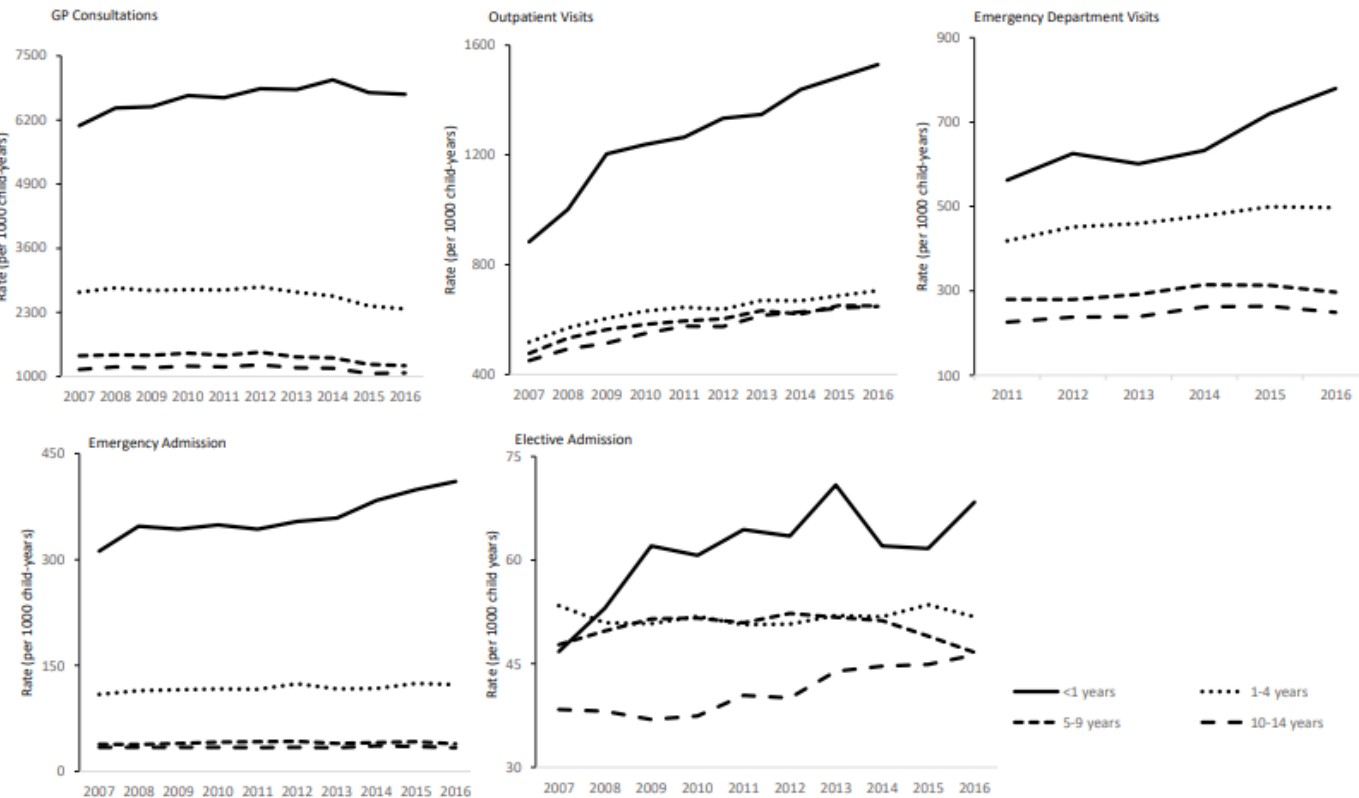

**Figure 2** Healthcare use in children by developmental age groups. GP, general practitioner.

The introduction of 4 hour waiting target in ED in 2004 has been previously suggested as a contributing factor to increase in emergency admission rates.[6 28] However, less than 5% of children's admission breach the 4 hour waiting time target, so pressures within EDs are unlikely to explain the rising admission rate.[29] Similarly, there is little evidence that ED staff have lowered their threshold for admitting children in our findings since the attendance admission ratio remained around one in five across the decade. Thus, our findings and previous studies point to mounting pressures in the community and illustrate the remarkable ability of hospitals in England to respond to the needs of families with young children.

### Strengths and limitations
Our study is the first to report on trends in healthcare use by children for both primary and hospital care in the UK. This is among the largest studies to report healthcare use in children across the primary and secondary care interface in a health system with universal health coverage. The strengths of this study include its size, national coverage and representativeness of the UK population of both CPRD and HES data sources widely used for research. This reduces the possibility that our findings are due to chance. Few countries have health systems that enable the study of individual-level data on children and links between primary care and hospital data.

However, there are several limitations to consider. First, in common with many observational studies using administrative records, data quality is variable. Our data sets did not include consultations in some urgent care settings including NHS walk-in centres private providers. Use of HES A&E data was initiated in 2007/2008 but data quality and coverage did not improve until 2009. Hence we restricted our analysis of EDs to the period from 2011 onwards.[15] HES outpatient data were first available in 2003/2004 and have been validated for research from 2015/2016. The rate of outpatient attendances in 2015/2016 has been ascribed to fluctuation in data quality and thus has not been reported in this paper.[24] Second, we used a proxy for birthdate as CPRD only provides month of birth for children aged less than 15 years. This would overestimate the follow-up period and underestimate the rate. Third, children living in deprived areas are more likely to deregister from CPRD practices than children residing in more affluent postcodes. Hence, our findings may not be generalisable to these groups.[30] Finally, our findings may not be generalisable to certain patient groups missing from CPRD such as children using private healthcare services and homeless children.[11]

### Implications and future research
The large sustained rises we report in primary care, urgent and planned care in hospital in infants have important implications for primary care. If rising urgent care use and falling contact rates with GPs are evidence of difficulties in children's access to GP appointments, this is an important area for reform. It brings into question the continued selective financial incentive schemes for GPs to manage a rising multimorbidity burden in older patients. It is of concern that GPs are not universally trained in paediatrics and child health medicine as has been the

recommendation of the European Academy of Paediatrics recently.[31] Regardless, upward trends in planned and urgent healthcare use by children have major resource and cost implications for the NHS and we recommend the drivers and mediators of these changes warrant further research.[3 32 33] A further breakdown of data that includes ethnic background, index of multiple deprivations, repeat attendance, presenting problems and case mix would be immensely helpful in understanding these trends. The number of health visitors and midwives who are providing support has decreased in England and Wales in recent years as a result of funding cuts.[34] Accurate data on the effectiveness of this workforce in supporting parents are currently lacking but are recommended as an area for future research. Finally, there is scope to explore innovative approaches including parenting programmes and peer support for increasing parent's confidence and capacity in caring for their children.

## CONCLUSIONS

Over the past decade in England, general practice consultations have fallen for all children, except for infants. Children's use of hospital urgent and outpatient care has risen in all ages, especially infants. These changes may signify the need for better access to primary care and provision of specialist and community-based support for families with young children. Research to understand drivers and identify solutions to rising health service use is urgently needed.

**Contributors** SS and MB conceived and designed this study. JR prepared the data and carried out the statistical analysis overseen by EC and AB. JR, MB, EC, GG, AB, DSH and SS took part in interpreting the data for this study, commented on and helped to revise drafts of this paper and have approved the final version. SS is the guarantor.

**Funding** This article presents independent research commissioned by the National Institute for Health Research (NIHR) under the Applied Health Research (ARC) programme for North West London. SS is supported by the NIHR School for Public Health Research Programme. The Dr Foster Unit at Imperial is affiliated with the National Institute of Health Research (NIHR) Imperial Patient Safety Translational Research Centre. The NIHR Imperial Patient Safety Translational Centre is a partnership between the Imperial College Healthcare NHS Trust and Imperial College London. The Dr Foster Unit at Imperial College is grateful for support from the NIHR Biomedical Research Centre funding scheme.The views expressed in this publication are those of the author(s) and not necessarily those of the NHS, the NIHR or the Department of Health and Social Care.

**Disclaimer** The views expressed are those of the authors and not necessarily those of the NHS, the NIHR or the Department of Health and Social Care.

**Competing interests** None declared.

**Patient and public involvement** Patients and/or the public were not involved in the design, or conduct, or reporting, or dissemination plans of this research.

**Patient consent for publication** Not required.

**Ethics approval** We obtained ethical and scientific approval for the use of CPRD for our study from the Independent Scientific Advisory Committee Protocol number: 18_139.

**Provenance and peer review** Not commissioned; externally peer reviewed.

**Data availability statement** Data may be obtained from a third party and are not publicly available. Due to data sharing agreements, data are unavailable for sharing publicly.

**ORCID iDs**
Judith Ruzangi http://orcid.org/0000-0002-7602-1436
Sonia Saxena http://orcid.org/0000-0003-3787-2083

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
