## [Reviewer comments · BMJ Open]

ARTICLE DETAILS

TITLE (PROVISIONAL)	Trends in healthcare use in children aged less than 15 years; a population-based cohort study in England from 2007 to 2017
AUTHORS	Ruzangi, Judith; Blair, Mitch; Cecil, E; Greenfield, Geva; Bottle, Alex; Hargreaves, Dougal S; Saxena, Sonia

VERSION 1 – REVIEW

REVIEWER	Amanda Montalbano Children's Mercy Kansas City, USA
REVIEW RETURNED	27-Aug-2019

GENERAL COMMENTS	Retrospective analysis of national data on individual healthcare utilization patterns further stratified by age which shows increase in utilization for infants across all healthcare delivery site with largest increases in outpatient visits across all age groups, followed by ED utilization analyzed over half the duration of the study period in comparison to other healthcare sites. Major strengths: Large, longitudinal database with clear categorization methodology. Admissions were able to be categorized by planned and unplanned admission. Major weaknesses: Objective is to describe changes in utilization patterns for different sites of healthcare delivery, but three issues continued to haunt my review. 1) After-hours and urgent care facility visits were not included. This seems like a huge waste when they would be easily identified in the database and could have major implications on GP relationships. 2) Even after reading the supplementary materials, I could not understand the difference between GP consultations and outpatient visits. Were the outpatient visits to specialists? Are their GP visits not scheduled? 3) Why leave out primary care visits? These again help establish "relationship" with the GP, so would also have huge implications on whether or not patients and families feel they truly have a medical home with their GP. Background: Would be helpful to better delineate what changes occurred in 2004 and how those changes could theoretically lead to changes in healthcare seeking behavior. What were the major reforms? How did this lead to oversaturation. You touch on this in the discussion, but would have been helpful up front to allow me to connect the dots between intervention and expected outcome. Methods: I was surprised with CPRD including only 8% of the population, it may be more clear to state: "While only representing 8% of the UK population, age, sex, and ethnicity reflect
---

	demographics of the UK in general.” What about at the <15yo population? Is it more inclusive of this younger generation? My concerns about the outcomes are listed in the Major Weaknesses above. ED analysis: would have loved to see emergency admission rate comparisons between those referred by GP and those who self-referred. Results: Remove “urgent care” from all your prose about the emergency medical care that was received. By definition, you excluded urgent care visits from your outcomes. Discussion: Remove “urgent care” from all your prose about the emergency medical care that was received. By definition, you excluded urgent care visits from your outcomes. The third paragraph about the “rising needs in the child population” can not be extrapolated from the data you present here. There is no clinical information about coded diagnoses of the visits or co-morbidities of the patients analyzed in this data set. Unless you include this level of analysis in this manuscript, I would be hard pressed to allow this paragraph to remain pertinent to the paper. The following paragraph again talks of the “rise in urgent care use” which you do not actually include in your analyses. Urgent care is sought for concerns needing attention within 12 hours that do not require the advanced capabilities of the emergency room. You have analyzed the perceived need for emergency care. However, the discussion around the 2004 reforms and how they were perceived to impact childhood GP care is a salient point that should be emphasized sooner and remain a thread throughout the paper. The end of the paragraph mentions that ED admission rates have remained stable over the studied time period, but increased use of the ED suggests there is no change in the acuity of the presenting patients which argues against the previous statement that parents are seeking care in the inappropriate setting of the ED at higher rates than in the past. Rather, the data suggest that patients are just seeking care more frequently both appropriately and ‘inappropriately’. Limitations: I would also suggest mentioning how the <1 group is affected by your birthdate limitations. Also, how are newborn admissions (at time of birth) handled? Conclusions: Again, I urge you not to use “urgent” care as that is not analyzed here. Figure 1: While I appreciate the simple illustrative case that the vast majority of care is still GP, this seems a waste of a figure. I would suggest illustrating those non-GP visits that were referred vs self-referred. Figure 2 & 3: could be combined. Overall could be a widely cited paper and worth revising and resubmitting.
--	--

REVIEWER	Dinci Pennap ORISE Fellow Oak Ridge Associated University
-----------------	---

	U.S. Food and Drug Administration U.S.A
REVIEW RETURNED	29-Oct-2019

GENERAL COMMENTS	The authors of the manuscript 'Trends in healthcare use in children aged less than 15 years; a population-based cohort study in England from 2007-2017' describe trends in the use of primary care, planned hospitalization and urgent care by children and adolescents younger than 15 years old in England. They highlight an important area of clinical significance in a vulnerable population. However, some concerns are worth noting: Background Section:  1. Please edit for typos – '...crucial role in prioritizing referral on for specialist...' 2. Multiple comments are without references, please fix. The last 3 sentences of the first paragraph need references. 3. You refer to the major reforms in NHS primary care but do not explain what these reforms are or the exact aspects of the reforms that could be driving your hypothesis. Your background section could be stronger with a brief description of the reforms. Methods Section:  4. Study design: Please edit for typos – '...covers approximately 8% of that population.' 5. Population: Please edit for typos – '...01/04/2007 to 31/07/2017.' 6. Outcomes: There is no need to cite reference 10 in the first sentence. Please delete. 7. Analysis: Rates and proportions are not interchangeable. Please revise the sentence 'we calculated the conversion rate for emergency admission as the proportion of ED visits that led to emergency admission'. Results Section:  8. Figure 1 could be presented as a table or text only. A figure is unnecessary. Also, figure 1 title doesn't indicate that ED visits were assessed from 2011 to 2017. Please revise. 9. Urgent and emergency care: The increase in ED visits is very interesting. It would be informative to identify reasons for such visits (i.e. diagnostic groups in 2011 vs. 2017). 10. Planned Care: At the end of the first sentence, please reference Figure 2. Discussion Section:  11. Your findings are interesting. It is worth noting that GP consultation is still the leading category in all age groups. Please highlight that. 12. Comparison with previous studies: In the first sentence of the second paragraph, you say 'rising needs...and demand from families are most likely contributors....'. However, you did not identify these rising needs in your population. For example, you say 'preterm birth is increasing and is the single biggest cause of neonatal morbidity and mortality in the UK' --- But you have not told the readers how many of the infants in your cohort were born preterm. You go on to say 'Thus, rising emergency admissions in infants could be explained by a growing morbidity burden in infants compared with other children', yet, you have not demonstrated this fact in your cohort. A summary distribution of morbidity in your cohort could strengthen your discussion. 13. Please edit for typos – '...chronic conditions contribute substantially to total emergency admissions affecting.'
--

	14. Please provide a reference for the first sentence of the third paragraph. 15. The third paragraph provides some context for the NHS reform. As mentioned in number 3 above, this context could strengthen your background section.
--	--

VERSION 1 – AUTHOR RESPONSE

Reviewer 1

Major weaknesses: Objective is to describe changes in utilization patterns for different sites of healthcare delivery, but three issues continued to haunt my review. 1) After-hours and urgent care facility visits were not included. This seems like a huge waste when they would be easily identified in the database and could have major implications on GP relationships. 2) Even after reading the supplementary materials, I could not understand the difference between GP consultations and outpatient visits. Were the outpatient visits to specialists? Are their GP visits not scheduled? 3) Why leave out primary care visits? These again help establish “relationship” with the GP, so would also have huge implications on whether or not patients and families feel they truly have a medical home with their GP.

Response: Thank you for this important set of comments – we accept that our previous description may not have been clear to those not working in the English NHS and have revised the manuscript to explain more clearly. Specifically,

- i) **As described in the text, after hours visits to NHS GP practices and Emergency Departments were included in our analyses. However, visits to some urgent care facilities including NHS walk in centres and private providers are not included. We have acknowledged this in the discussion strengths and weaknesses section.**
- ii) **A GP consultation is with a general practitioner (family physician) in the community, whereas an outpatient visit is with a hospital-based specialist, for children this is usually a paediatrician. Consultations with specialists require onwards referral from GPs.**

This is stated in the background section:

‘GPs play a crucial role in prioritising onwards referral for specialist care in hospital and outpatient settings.’ We have also clarified this distinction between primary and hospital care in the abstract methods. To summarise this and recent policy change in NHS health care, a definition of outpatients and GP consultation as well as reforms has been included in Table 1.

The majority of primary care visits are face to face with the GPs (1.6 per child per year in 2016) so are already included in these analyses. In 2016, the telephone contact rates were 48.3 per 1000 child-years and home visit rates 0.341 per 1000 child-years. Primary care contacts with nurses and other professionals represent a relatively small proportion of primary care contacts for children and young people; analyses of these contacts was beyond the scope of this article.

Background: Would be helpful to better delineate what changes occurred in 2004 and how those changes could theoretically lead to changes in healthcare seeking behavior. What were the major reforms? How did this lead to oversaturation. You touch on this in the discussion, but would have been helpful up front to allow me to connect the dots between intervention and expected outcome.

Response: A summary of the 2004 NHS policy reforms appears in the Background section and has been reported extensively by us and others. Essentially this was an opting out of ‘out of hours’ responsibility, adopted by almost all GPs, and introduction of tighter targets for waiting times in emergency care settings. Concurrent with these changes was a new pay for performance scheme that remunerated GPs for work previously done by hospital specialists. This scheme expanded GP workload for chronic disease management in adults and impacted appointment availability for those with non-incentivised conditions.

We have added an at-a-glance description of these key reforms in Table 1. Additional information on these reforms is available in references 1, 4-7 in the revised manuscript.

Methods: I was surprised with CPRD including only 8% of the population, it may be more clear to state: “While only representing 8% of the UK population, age, sex, and ethnicity reflect demographics of the UK in general.” What about at the <15yo population? Is it more inclusive of this younger generation?

Response: Thank you, this sentence has been revised accordingly. As described in the text and references, the age distribution of the CPRD sample is broadly representative of the England population.

My concerns about the outcomes are listed in the Major Weaknesses above.

ED analysis: would have loved to see emergency admission rate comparisons between those referred by GP and those who self-referred.

Response: We appreciate this interesting suggestion on comparing children who were referred by GP and those who self-referred. We have previously reported that 35-40% of hospitalised children under 15 years are referred by GPs although this has been falling by around 1% per year in recent years. This data is captured more accurately as it required a written referral if the child sees a GP beforehand (7). However, the accuracy of GP referred ED visits is a complex area to study because some referrals will be by telephone. This detail was however beyond the scope of our paper.

Results: Remove “urgent care” from all your prose about the emergency medical care that was received. By definition, you excluded urgent care visits from your outcomes.

Discussion: Remove “urgent care” from all your prose about the emergency medical care that was received. By definition, you excluded urgent care visits from your outcomes.

Conclusions: Again, I urge you not to use “urgent” care as that is not analyzed here.

Response: Thank you for this thoughtful comment which has highlighted to us that readers from outside the UK may interpret urgent care differently. Therefore, in the revised manuscript, we have included a definition of urgent care as used in our manuscript within Table 1 in the Background Section to avoid confusion.

The third paragraph about the “rising needs in the child population” can not be extrapolated from the data you present here. There is no clinical information about coded diagnoses of the visits or co-morbidities of the patients analyzed in this data set. Unless you include this level of analysis in this manuscript, I would be hard pressed to allow this paragraph to remain pertinent to the paper.

The following paragraph again talks of the “rise in urgent care use” which you do not actually include in your analyses. Urgent care is sought for concerns needing attention within 12 hours that do not

require the advanced capabilities of the emergency room. You have analyzed the perceived need for emergency care. However, the discussion around the 2004 reforms and how they were perceived to impact childhood GP care is a salient point that should be emphasized sooner and remain a thread throughout the paper. The end of the paragraph mentions that ED admission rates have remained stable over the studied time period, but increased use of the ED suggests there is no change in the acuity of the presenting patients which argues against the previous statement that parents are seeking care in the inappropriate setting of the ED at higher rates than in the past. Rather, the data suggest that patients are just seeking care more frequently both appropriately and 'inappropriately'.

Response: Thank you for this thoughtful comment. Emergency department diagnostic codes are generally not used due to poor data quality of that variable. Additionally, investigating ICD-10 codes that are used for admitted patients was outside the scope of this paper. This will be studied more closely in the next paper from our group.

Please note the previous response regarding different use of the term 'urgent care' and our clarification of the use of this term in this article. More generally, our data provide evidence of increasing demand for ED care, especially in babies. We agree that it is unclear to what extent this reflects increasing need, lack of GP appointments and other community services, or other influences on healthcare seeking behaviour. Previous studies have highlighted some examples where there is clear evidence of increasing need – for example ex premature babies – but we agree that the overall relationship between increased demand and increased need is complex and cannot be fully investigated within the data we present in the paper. We also note other specific areas where the relationship between increased demand for services and increased prevalence of medical disorders is disputed or unclear (1).

Limitations: I would also suggest mentioning how the <1 group is affected by your birthdate limitations. Also, how are newborn admissions (at time of birth) handled?

Response: We greatly appreciate this constructive comment. Actually, the follow up time for under 1s would be overestimated because of the date ascribed. The maximum overestimation per child would be 1 month. In which case, the denominator is an overestimate and the rate is therefore underestimated. In the revised manuscript, we mentioned how the under 1 group was affected by the birthdate limitations. New-born admissions codes were excluded.

Figure 1: While I appreciate the simple illustrative case that the vast majority of care is still GP, this seems a waste of a figure. I would suggest illustrating those non-GP visits that were referred vs self-referred. Figure 2 & 3: could be combined.

Response: We are grateful to the reviewer for these suggestions on the figures on our manuscript.

In the revised manuscript, we have removed figure 1 completely as it fully described by the first paragraph in the Results section as follows; "Overall, 1,484,455 children from 408 practices contributed to this 10-year cohort study. From 1 April 2007 to 31 March 2017, there were 7,604,024 GP consultations, 981,684 ED visits, 287,719 emergency admissions, 194,034 elective admissions, 2,253,533 outpatient visits among children aged less than 15 years in our population."

In the revised manuscript, Figure 2 and 3 were not combined as the scales of the charts are different. By having them as separate graph one can easily see the trends over time.

As above, we have previously reported that 35-40% of hospitalised children under 15 years are referred by GPs although this has been falling by around 1% per year in recent years. This data is captured more accurately as it required a written referral if the child sees a GP beforehand (7). However, the accuracy of GP referred ED visits is a complex area to study because some referrals will be by telephone.

Reviewer 2

Background Section:

1. Please edit for typos – ‘...crucial role in prioritizing referral on for specialist...’

Response: We greatly appreciate the reviewer’s careful review. BMJ Open allows a mixture of English and American spelling, depending on the provenance and main target audience of the article. As we believe this target audience to be in the UK, we have left the spelling as English (UK).

2. Multiple comments are without references, please fix. The last 3 sentences of the first paragraph need references.

Response: We appreciate the reviewer’s helpful comment so have included appropriate references for the last 3 sentences of the first paragraph.

3. You refer to the major reforms in NHS primary care but do not explain what these reforms are or the exact aspects of the reforms that could be driving your hypothesis. Your background section could be stronger with a brief description of the reforms.

Response: We appreciate the reviewer’s thoughtful comment on the description of reforms. Our previous description may not have been clear to those not working in the English NHS and have revised the manuscript to explain more clearly. A description of reforms has been included in table format in the background section (see Table 1).

Methods Section:

4. Study design: Please edit for typos – ‘...covers approximately 8% of that population.’

5. Population: Please edit for typos – ‘...01/04/2007 to 31/07/2017.’

6. Outcomes: There is no need to cite reference 10 in the first sentence. Please delete.

7. Analysis: Rates and proportions are not interchangeable. Please revise the sentence ‘we calculated the conversion rate for emergency admission as the proportion of ED visits that led to emergency admission’.

Response: We thank the reviewer’s for these typos suggestions. The typos have been revised

-Study Design section: “While only representing 8% of the UK population”

-Population section: “01/04/2007 to 31/07/2017”

-Reference 10 in the first sentence in the outcomes section has been deleted as suggested.

- Analysis Section under Methods “conversion rate” amended to “conversion proportion”.

Results Section:

8. *Figure 1 could be presented as a table or text only. A figure is unnecessary. Also, figure 1 title doesn't indicate that ED visits were assessed from 2011 to 2017. Please revise.*

Response: We thank the Reviewer's for this thoughtful suggestion. In the revised manuscript, we have removed figure 1 completely as it fully described by the first paragraph in the Results section as follows; “Overall, 1,484,455 children from 408 practices contributed to this 10-year cohort study. From 1 April 2007 to 31 March 2017, there were 7,604,024 GP consultations, 981,684 ED visits, 287,719 emergency admissions, 194,034 elective admissions, 2,253,533 outpatient visits among children aged less than 15 years in our population.”

9. *Urgent and emergency care: The increase in ED visits is very interesting. It would be informative to identify reasons for such visits (i.e. diagnostic groups in 2011 vs. 2017).*

Response: As in our response to reviewer 1, we greatly appreciate reviewer 2's suggestion as we also think this is an interesting question here. We explored diagnostic codes for those visiting A&E, however, emergency department diagnostic codes are generally not used due to insufficient data quality.

10. *Planned Care: At the end of the first sentence, please reference Figure 2.*

Response: We thank the Reviewer's for this thoughtful suggestion. We have now referenced the appropriate figure at the end of the first sentence in Planned Care in the Results Section.

Discussion Section:

11. *Your findings are interesting. It is worth noting that GP consultation is still the leading category in all age groups. Please highlight that.*

12. *Comparison with previous studies: In the first sentence of the second paragraph, you say ‘rising needs...and demand from families are most likely contributors...’. However, you did not identify these rising needs in your population. For example, you say ‘preterm birth is increasing and is the single biggest cause of neonatal morbidity and mortality in the UK’ --- But you have not told the readers how many of the infants in your cohort were born preterm. You go on to say ‘Thus, rising emergency admissions in infants could be explained by a growing morbidity burden in infants compared with other children’, yet, you have not demonstrated this fact in your cohort. A summary distribution of morbidity in your cohort could strengthen your discussion.*

Response: We appreciate the reviewer's thoughtful and constructive comment.

As a result of this comment we have highlighted that “GP consultations are the major form of contact with healthcare across all age groups” in the first paragraph in the Discussion Section.

It would have been useful to show a distribution of morbidity in our cohort to strengthen our discussion as suggested. Unfortunately, ED codes are not used so this was not possible. As stated above we have previously explored reasons for hospital admission in children but a detailed exploration of ICD-10 codes in this paper was beyond the scope of this paper. There is evidence from smaller studies of increasing demand for ED care, especially in babies. It is unclear to what extent this reflects increasing need, lack of GP appointments and other community services, or other influences on healthcare seeking behaviour. There are some

examples where there is clear evidence of increasing need - example ex premature babies - but can't investigate the population attributable fraction (PAF) of increased demand due to ex premature babies in our paper. In other areas, relationship between increased demand for services and increased prevalence of medical disorders is dispute (1).

13. Please edit for typos – ‘...chronic conditions contribute substantially to total emergency admissions affecting.’

Response: We appreciate the reviewer’s helpful comment. In the revised version, we have deleted the word “affecting” from this sentence -Discussion Section under Comparison with Previous Studies.

14. Please provide a reference for the first sentence of the third paragraph.

Response: Thank you for highlighting this. In the revision document, we have referenced the appropriate report.

15. The third paragraph provides some context for the NHS reform. As mentioned in number 3 above, this context could strengthen your background section.

Response: We greatly appreciate the reviewer’s positive comment. In the revised version of the manuscript, we have included a description of reforms in Table 1 in the Background Section.

VERSION 2 – REVIEW

REVIEWER	Amanda Montalbano Children's Mercy Kansas City, USA
REVIEW RETURNED	15-Jan-2020

GENERAL COMMENTS	The addition of the summary box in Table 1 was an excellent improvement to the readability of the manuscript. I have attached further comments/suggestions in the document below. Most are small considerations, but I would urge the authors to reorganize the discussion to clarify and emphasize the poignant findings in the data in relation to the reforms from 2004. I appreciate the authors’ thorough responses to both reviewers’ critiques and suggestions. The biggest improvement was the inclusion of Table 1’s summary of reform changes and definitions that clarified the authors’ line of inquiry. I have a few follow up suggestions. 1. The analysis section states, “We calculated the proportion of children with an ED visit who selfreferred. We calculated the conversion proportion for emergency admission as the proportion of ED visits that led to emergency admissions.” a. A previous suggestion to compare the proportions of emergency admissions between those who self-referred to the ED vs those would were directed to present to the ED was described as
--

	a detail beyond the scope of this paper. I am unclear why it is beyond the scope of your paper if you have both the proportion of children who self-refer to the ED AND 2. The proportion of ED visits that become emergency admissions. Either remove the self-refer analyses from the manuscript (as they are not reported in figure form) or do something with the data such as: % self-referral by age group if the overall percent of self-referral remained stable over time or emergency admission proportion for self-referrals as mentioned above. b. How did you calculate the emergency admission proportions from 2007-2010 if you only have ED visit data from 2011 on? 2. If your study population is children less than 15 years of age, you only need to state that in the methods. It is distracting when it is repeatedly inserted in the analyses and results. 3. The prose of the results may be an easier read if you report the “X% [increase/decrease] by year or from baseline” to decrease the frequency of roman numbers bombarding the readers and leave the actual numbers to be investigated by the readers in Table 2. 4. Consider reorganizing the rows in Table 2 to follow a grouping of outpatient utilization (GP consults, Outpatient visits, ED visits) then admissions (Elective, Emergency). Change the column headers (Baseline, 2017, % change) then note the baseline year difference in the footer. Also consider adding a column for overall/total population combined. 5. I appreciate the scales on the Figures are different in each panel in Figure 2 and 3, and this is one reason I will continue to advocate that you should consider collapsing some of the panels. Because the scale is almost 10-fold different between GP consults and ED visits, yet they are side by side, it makes the illusion that the increase in ED visits rapidly outpaces the decrease of GP consults when in reality the ED visit rate increased around 5% and GP consults decreased around 14%. My suggestions: a. Collapse figure 1 into 2 panels: 1) GP consults, ED visits, outpatient visits – even if you have 2 scales, one each Y-axis (one for GP the other for outpatient/ED), 2) ED admissions and elective admissions (y-axis could easily run from 40-90). b. I think figure 2 could stay as separate panels for each category as the purpose is to compare the age groups within that category and not across categories; however you may want to reorganize the order in which they appear to mirror the order presented in the revised Table 2. 6. I would continue to advocate to move up the summary statement, “The 2004 primary care healthcare reforms focused heavily on improving....this reduced access [that] may explain the rising emergency department visits” to the beginning of your discussion as I see this as the basis of the paper. Explore
--	---

	overall patterns outpatient utilization. Next highlight key finding in infant population, exploring the increase in premature births contributing to a more complicated neonatal cohort that may be beyond the GPs' comfort, leading to increase specialist and ED use in this population. Then discuss the pattern of stable elective and emergency admissions (except for those complicated neonates raised in last section) even with the 4-hour reforms applying pressure to disposition patients. Again, would be great if you could discuss patterns seen between the cohort of those who self-referred to the ED vs others as this would help support the 'demand of families' you continue to propose throughout the discussion.
--	---

VERSION 2 – AUTHOR RESPONSE

Responses to comments

I appreciate the authors' thorough responses to both reviewers' critiques and suggestions. The biggest improvement was the inclusion of Table 1's summary of reform changes and definitions that clarified the authors' line of inquiry. I have a few follow up suggestions.

1. The analysis section states, "We calculated the proportion of children with an ED visit who self-referred. We calculated the conversion proportion for emergency admission as the proportion of ED visits that led to emergency admissions."

a. A previous suggestion to compare the proportions of emergency admissions between those who self-referred to the ED vs those would were directed to present to the ED was described as a detail beyond the scope of this paper. I am unclear why it is beyond the scope of your paper if you have both the proportion of children who self-refer to the ED AND 2. The proportion of ED visits that become emergency admissions. Either remove the self-refer analyses from the manuscript (as they are not reported in figure form) or do something with the data such as: % self-referral by age group if the overall percent of self-referral remained stable over time or emergency admission proportion for self-referrals as mentioned above.

Thank you for this insightful comment. Analysis of self-referred ED attendances are not central to the stated aims of our paper, as suggested, we have now removed the self-referral result in the analyses and result.

b. How did you calculate the emergency admission proportions from 2007-2010 if you only have ED visit data from 2011 on?

The conversion proportion refers to the proportion of attendees who are subsequently admitted as in-patients. The conversion proportion was calculated from 2011 since that is the data of ED visits of good quality. We didn't report these proportion from 2007 because of the ED visit data quality which is reflected by the proportions in 2007 as well.

However, we have made clear in the Methods section under Analyses that these proportions were from 2011/2012.

2. If your study population is children less than 15 years of age, you only need to state that in the methods. It is distracting when it is repeatedly inserted in the analyses and results.

We are grateful for this practical comment. We have now removed the repeating phrase of children aged less than 15 years appropriately in the results and analyses sections.

3. The prose of the results may be an easier read if you report the “X% [increase/decrease] by year or from baseline” to decrease the frequency of roman numbers bombarding the readers and leave the actual numbers to be investigated by the readers in Table 2.

We thank the reviewer for this practical suggestion. We have now changed the results section to read better by removing the from/to roman numbers for rates and left the X%/year increase and decrease.

4. Consider reorganizing the rows in Table 2 to follow a grouping of outpatient utilization (GP consults, Outpatient visits, ED visits) then admissions (Elective, Emergency). Change the column headers (Baseline, 2017, % change) then note the baseline year difference in the footer. Also consider adding a column for overall/total population combined.

We appreciate this useful suggestion. Rows in Table 2 have now been reorganised to follow recommended order. They now start with GP consultations, Outpatient visits, ED visits followed by Elective and Emergency admissions respectively. Column headers have also been changed from “2007/08, 2016/17, % change” to read “Baseline, 2017, % change” as can be seen on the manuscript on Table 2. Adding another column to the table makes it hard to read as it is already in landscape. The total population has been reported in the results section.

5. I appreciate the scales on the Figures are different in each panel in Figure 2 and 3, and this is one reason I will continue to advocate that you should consider collapsing some of the panels. Because the scale is almost 10-fold different between GP consults and ED visits, yet they are side by side, it makes the illusion that the increase in ED visits rapidly outpaces the decrease of GP consults when in reality the ED visit rate increased around 5% and GP consults decreased around 14%. My suggestions:

a. Collapse figure 1 into 2 panels: 1) GP consults, ED visits, outpatient visits – even if you have 2 scales, one each Y-axis (one for GP the other for outpatient/ED), 2) ED admissions and elective admissions (y-axis could easily run from 40-90).

We appreciate this practical suggestion and have made changes accordingly. We have now changed Figure 1 to only have 2 panels as seen on Figure 1.

b. I think figure 2 could stay as separate panels for each category as the purpose is to compare the age groups within that category and not across categories; however you may want to reorganize the order in which they appear to mirror the order presented in the revised Table 2.

Figure 2 has now changed to landscape mode to accommodate the changes requested by the reviewer. The order has been changed to mirror the revised Table 2.

6. I would continue to advocate to move up the summary statement, “The 2004 primary care healthcare reforms focused heavily on improving....this reduced access [that] may explain the rising emergency department visits” to the beginning of your discussion as I see this as the basis of the paper. Explore overall patterns outpatient utilization. Next highlight key finding in infant population, exploring the increase in premature births contributing to a more complicated neonatal cohort that may be beyond the GPs’ comfort, leading to increase specialist and ED use in this population. Then discuss the pattern of stable elective and emergency admissions (except for those complicated neonates raised in last section) even with the 4-hour reforms applying pressure to disposition patients. Again, would be great if you could discuss patterns seen between the cohort of those who self-referred to the ED vs others as this would help support the ‘demand of families’ you continue to propose throughout the discussion.

The discussion section has now been revised to follow the reviewer’s suggestion.

VERSION 3 – REVIEW

REVIEWER	Amanda Montalbano Children's Mercy Kansas City, USA
REVIEW RETURNED	20-Mar-2020
GENERAL COMMENTS	Wonderful update to the manuscript after revisions. Much easier read and flow. Will be great addition to the literature.